# Using unsupervised learning to detect broken symmetries, with relevance to searches for parity violation in nature.

Christopher G. Lester[1] and Rupert Tombs[1]

[1] Cavendish Laboratory, University of Cambridge, CB3 0HE, United Kingdom

**Reviewed on OpenReview:** `https://openreview.net/forum?id=QFJ3gtbwHR`

## Abstract

Testing whether data breaks symmetries of interest can be important to many fields. This paper describes a simple way that machine learning algorithms (whose outputs have been appropriately symmetrised) can be used to detect symmetry breaking. The original motivation for the paper was an important question in Particle Physics: "Is parity violated at the LHC in some way that no-one has anticipated?" and so we illustrate the main idea with an example strongly related to that question. However, in order that the key ideas be accessible to readers who are not particle physicists but who are interesting in symmetry breaking, we choose to illustrate the method/approach with a 'toy' example which places a simple discrete source of symmetry breaking (the handedness of human handwriting) within a idealised particle-physics-like context. Readers interested in seeing extensions to continuous symmetries, non-ideal environments or more realistic particle-physics contexts are provided with links to separate papers which delve into such details.

## 1 Introduction

On the 18[th] June 2021, after more than a decade studying collision data from the Large Hadron Collider (LHC), the ATLAS Collaboration (The ATLAS Collaboration, 2008) submitted for publication its 1000[th] paper. Despite that huge body of work, not a single one of those papers (and not a single one of the papers of any other LHC collaboration then or since) has yet looked for new mechanisms of overt[1] parity violation.

The main reason for this omission is that there are few theoretical models which predict forms of overt parity violation that the LHC could see given its unpolarised collisions, and so without a steer from theory to guide the construction of specific analysis strategies, a view seems to have formed that progress in this area is either a waste of time or simply impossible to perform in an efficient manner.

The purpose of this paper is to dispel that myth – but to do so in a toy/idealised environment so that the paper is more accessible to readers in other fields who might also be looking to detect symmetry violation. By using the handedness of human handwriting as our source of symmetry breaking we intend to emphasise that the method does not require potential sources of symmetry violation to have been known about in advance and simulated — the method processes only recorded data without additional simulation input. Readers who would like to see the framework put into use in 'real' particle physics context are encouraged to read a more detailed sister paper Lester et al. (2022b). Extensions to continuous symmetries and non-ideal environments can be found in a different sister paper Tombs & Lester (2022). Eschewing such details, the present paper hopes that its decision to motivate the key idea via a test for a discrete broken symmetry (parity) in an idealised environment will enlarge the potential readership.

---

[1] The Standard Model of Particle Physics has been known to violate parity since the 1950s. The discovery by Wu et al. (1957) relied on the observation that beta particles emitted from certain nuclei were found to be preferentially emitted anti-parallel to the initial state nuclear spins of those nuclei – an observation which *only* admits explanations which violate parity. It was not necessary for the underlying (parity violating) *mechanism* leading to those observations to have been known or understood for

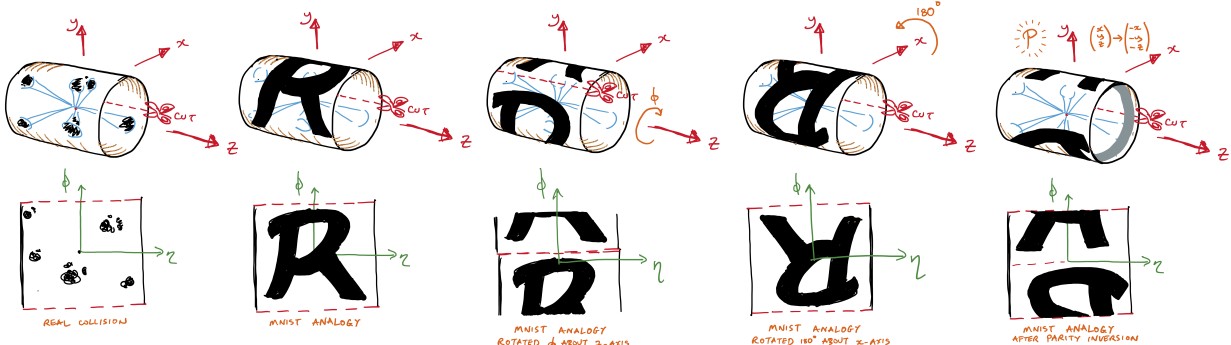

Figure 1: This figure aims to show how transformations of a 'real' three-dimensional collider calorimeter (top row) correspond to induced transformations in the unrolled two-dimensional $(\eta, \phi)$-space in which those deposits are often represented. The first column shows deposits that would be typical of a five jet event. The second column shows an easier-to-visualise set of deposits resembling a letter '$\mathbf{R}$'. The other three columns show how the deposits of the first '$\mathbf{R}$' would move around if affected by (i) an arbitrary rotation about the $z$-axis, (ii) a rotation of 180 degrees about the $x$-axis and (iii) a parity inversion.

## 2  Method and Principles

The symmetry we use to illustrate the method is 'parity'[2] – the symmetry which is its own inverse and which interchanges any object $x$ with its mirror image $Px$. In physics and chemistry parity is usually thought of as a spatial operation acting on three dimensional structures $x$ such as collider events or molecules. Parities can, however, act on many other things.[3] In our illustration each object $x$ will be a small image containing information about a collision. A function $f$ mapping the input space to one on which a concept of negation exists will be termed 'parity odd' if and only if $f(x) = -f(Px)$ for all $x$.

The core principles underlying the method which this paper proposes for making for general parity-violation detections are: (i) that (by definition!) a parity-odd function $f(x)$ can only have an asymmetric distribution under parity if the dataset on which it has been evaluated contains some elements $x_1$, $x_2$, ... which differ in frequency from their parity-flipped partners $Px_1$, $Px_2$, ...; and that therefore (ii) the observation of any statistically significant asymmetry in any parity-odd variable when evaluated on a real dataset is bona fide evidence for parity violation in that dataset; and so (iii) the goal of the LHC collaborations [glossing over some important real-world considerations which are mentioned later] should be to create parity-odd functions $f(x)$ which, after optimisation for asymmetry on a 'training' part of a real dataset, are then tested for asymmetry on other independent portions of the same dataset.

Any function $g(x)$ can be trivially made into a parity-odd function $f(x)$ by antisymmetrisation: $f(x) \equiv g(x) - g(Px)$, and every parity-odd function $f(x)$ can be so-constructed. Thus goal of 'principle (iii)' above can be met by using any of the latest tools of machine learning to provide diverse functions $g(x)$ which can be optimised for the asymmetries they evoke in $f(x)$ on the training portion of a dataset of interest. The power in the method comes from the marriage between (a) the capacity of machine learning algorithms to eke-out the best $g(x)$ for a given domain or dataset, and (b) the symmetry used to build $f(x)$ from $g(x)$ (imposed by hand rather than learned!) which guarantees that any asymmetry seen in a test-set evaluation is interesting.

---

that discovery of parity violation to have been declared. It is signs such as those – signs which can *only* be explained away as parity violating (as opposed to being effects which *could* be parity violating but which could also, in principle, be accounted for by alternative parity conserving explanations) – which we call signs of 'overt' parity violation.

[2]More general symmetries (including continuous and non-binary symmetries) are dealt with in a more complex sister paper Tombs & Lester (2022).

[3]Arithmetic negation on the real line is perhaps one of the simplest parities imaginable.

For tests of parity in particle physics this process is general[4], avoids all first-order use of Monte Carlo, and (like the original discovery of parity violation by Wu et al. (1957)) requires no knowledge of the parity violation mechanism to reach its conclusion.

We later choose to *illustrate* the above principle in the simplest way we can imagine:

- by creating a neural-net[5] $g(x)$ which acts on collisions $x$,

- by (anti)symmetrising the output of $g(x)$ into a parity-odd function $f(x)$ using $f(x) = g(x) - g(Px)$, and then

- training the net $g(x)$ to maximise the significance of the distance between the mean of $f(x)$ (when evaluated on test dataset) and zero.[6]

This illustration relies on the fact that a symmetric distribution with a mean will have a mean of zero, and so statistically significant non-zeroness in the mean is a measure of asymmetry.[7]

**Unusual features of this core idea (as distinct from its implementation in our particular illustration)**

It must be emphasised how different this proposed use of machine learning is to that most frequently seen in particle physics. There are many ways one could illustrate this, but perhaps the most easy one to highlight is that the training step is (technically, though not in any practical sense!) optional. If the training step were entirely omitted, the net $g(x)$ would be in a random state but the resulting $f(x)$ would still be parity-odd. If the evaluation of this $f(x)$ on the test set were to yield a statistically significant asymmetry by a measure fixed in advance (e.g. a non-zero mean) then *that asymmetry would be valid evidence for parity violation*! In practice, of course, the training step is very important. Training cannot be ignored. Without training there is almost no chance that the net $g(x)$ will have the right structure to generate an asymmetry in $f(x)$, even on a parity-violating dataset. But the point we wish to stress is that the training process exists not to make the discovery of parity-violation *possible*, but only to make it much more likely that a discovery of parity-violation can be made rather than missed.

For related reasons 'over-training'[8] does not have the same negative overtones it would normally have[9], at least for experimental physicists whose greatest concern is avoiding a false positive (a claimed discovery of a symmetry violation when it does not really exist). The possibility of over-training still exists but it cannot lead to an accidental mis-discovery of parity-violation as there is simply no concept of 'training that can mislead', only of 'more useful' or 'less useful' training. For a discovery of parity violation all that matters is that the appropriate net symmetrisation ($f(x)$) manages to show a statistically significant asymmetry on the test data. If it does, it's a good net, however it got there. The worst that can happen is that in training the net attends to entirely the wrong things (all unrelated to parity) and so does not know how to generate an asymmetry in the test-set (or a mean away from zero in our illustrative example) even if a better-trained net could have done so. For example: if the training resulted in $g(x)$ evaluating to a constant '$s$' on every element of the training set, and ended up evaluating to the constant '$t$' on everything else, then the possible output of $f(x)$ would be in the set $S = \{t - s, 0, s - t\}$. In the case that $s = t$, this set would contain only a single element ($S = \{0\}$) and so $f(x)$ would be incapable of showing an asymmetry on *any* dataset, making false positives impossible. In the case that $s \neq t$, the set $S$ would contain three elements: $t - s$, 0 and $s - t$.

---

[4]A lengthy proof of sufficiency regarding the use of parity-odd variables is presented in a sister paper (Lester, 2021a).

[5]It is a shame that we have not used a generative adversarial network, for if we had we could have called our paper a tabloid newspaper title 'Stressed GANs snag desserts'. Such a title would have been ideal given its symmetry properties.

[6]As will be seen later, this is a mean which has been suitably normalised so as to prevent the network gaining by (say) simply multiplying the net final output by a number greater than one.

[7]Note that our decision to use a neural-net rather than some other function approximator, or our decision to optimise for deviations of the mean of that parity-odd variable from zero rather than to optimise for some fancier loss function, are all unashamedly likely to be non-optimal. The sole aim of the paper is not to show which fancy loss function or network is best, but is rather to demonstrate that general searches for parity violation can be accomplished *at all* and with relative ease.

[8]During review it was suggested that 'over-fitting' may have be a more standard term in the ML field for what we have referred to as 'over-training'. As non-specialists we would happily use either term.

[9]A common reaction the authors have experienced when describing their proposal to newcomers is 'My gosh! Are you not proposing a mechanism for maximising over-training problems!?'

In such a case $f(x)$ could show an asymmetry on a test set – and (as mentioned already) any statistically significant asymmetry in the test set is valid evidence for symmetry-violation.

We note that the proposal is also unusual in that the test and training datasets are identically distributed, consist entirely of data (no Monte Carlo), and are unlabelled — and so are immediately usable in high luminosity and intense environments where simulation can be both complex and costly.

**Benefits for particle physicists, even if nature does not violate parity in a new way**

No detector is perfectly aligned or calibrated or without regions where acceptance can vary. Consequently evidence of what we have called 'parity violation' cannot be pinned immediately at nature's door. On the contrary, it is quite possible that the biggest sources of parity violation could be mis-alignments and mis-calibrations in the detectors used. This is not really a disadvantage of the method, though. Instead it is a sign that if you can get a significant asymmetry on the test-sample, then you know immediately that **either** you don't understand something about your detector, **or** you have an interesting discovery. Both are equally important things to know, so a signal is a win-win outcome even if the source of the parity-violation is not immediately obvious. Whether the same win-win situation occurs in other fields is less clear. For example, a biologist using a net to detect parity violation in pictures of snails (shells are handed) may not be pleased if the parity violation which has been detected in those images is later found to have been caused by things in the background of each image (e.g. road signs, leaves, cars) rather than being due to the snails themselves.

**Removing real-world complications**

Real detectors and experiments are, of course, imperfect. Not all parts work with perfect efficiency, and acceptance does not extend uniformly into all directions. A more complex sister paper (Tombs & Lester, 2022) address issues arising from those concerns, including: how to spot symmetry violation in a detector which itself violates that symmetry, and how to generalise the symmetry-violation discovery process to general symmetries (including continuous symmetries) rather than the narrow case of parity discussed herein. The current paper, however, deliberately eschews those details as the intention here is to provide a simple and pedagogically useful illustration of the key message, free from other distractions.

## 3 Data for an illustrative example

**Visualising deposits in the 'MNIST Calorimeter'**

Calorimetric deposits in idealised versions of detectors like ATLAS or CMS are commonly represented as functions on a cylinder about the beam axis as depicted in the upper left corner of Figure 1. This cylinder may be opened out into a rectangle with pseudorapidity $\eta$ on one axis and azimuthal angle $\phi$ on the other as depicted in the lower left corner of the same figure. In this flattened view the upper and lower edges are identified with each other.

Real calorimetric deposits are blotchy and messy, and this makes it hard for humans to picture what they might look like when rotated or mirrored. Since the goals of this paper are pedagogical, we choose to imagine a strange world in which calorimetric deposits look just like hand-written letters of the alphabet. For example, the second column of Figure 1 imagines what would happen if calorimeter deposits happened to have landed in locations which (once unfolded) would resemble the capital letter '**R**'. Making use of this visual simplification, the other columns of Figure 1 illustrate the effect that different transformations would have on the first '**R**'.

**The LHC datasets**

A real analysis would use real data not Monte Carlo simulations. Unfortunately, as there is no real detector that produces energy deposits which look like letters, we must generate some toy data of our own for our pedagogical purposes. There is a ready supply of hand-written numerals in the MNIST dataset (LeCun & Cortes, 2010) so we implement a generator that draws its calorimeter deposits from there. We call these

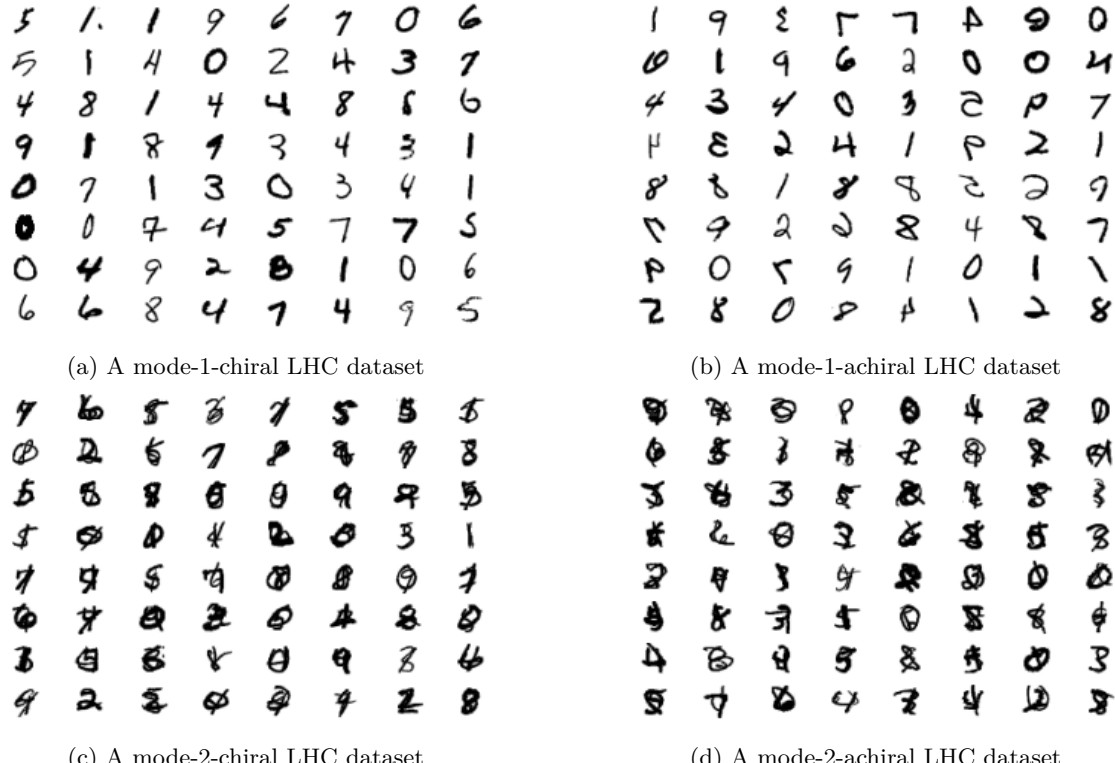

(a) A mode-1-chiral LHC dataset

(b) A mode-1-achiral LHC dataset

(c) A mode-2-chiral LHC dataset

(d) A mode-2-achiral LHC dataset

Figure 2: Four types of LHC dataset are shown. Further descriptions can be found in the body-text.

*Large* datasets of *H*andwritten *C*alorimeter digits – or 'LHC datasets' for short. We produce LHC datasets in four modes as shown in Figure 2 and defined as follows:

- **mode-1-chiral** : These are normal MNIST digits. These samples are chiral because humans write hand-written digits with handed-appendages called hands. The word chiral even come from the ancient Greek word for hand: $\chi\epsilon\iota\rho$;

- **mode-1-achiral** : These are MNIST digits which have been reversed left-to-right with probability 50%; The symmetrisation process makes these samples achiral.

- **mode-2-chiral** : These are random pairs of normal MNIST digits superimposed on each other. They inherit their chirality from their constituent digits.

- **mode-2-achiral** : These are random pairs of normal MNIST superimposed on each other with one member of the pair forwards and one reversed. The symmetrisation process makes these samples achiral.

Mode-1-achiral acts as a control for mode-1-chiral, and mode-2-achiral acts as a control for mode-2-chiral. The only purpose of mode-2 is to illustrate that the analysis is not dependent on there being only a small number of 'types' of deposit — in this case ten as there are ten sorts of digits. Both types of Mode-2 therefore act as a form of control against the possibility that mode-1 is too simple to be a good proxy for real and highly variable calorimeter data.

**Removing remaining differences between real calorimeter data and MNIST derived calorimeter analogues**

MNIST digits are 'oriented' because the humans who originally generated them were instructed to write them in boxes the conventional way up.

Real jet data, in contrast, is not oriented: if energy deposits were to appear in a configuration resembling a capital '**R**' then the deposits would be as likely to arrive in any of the configurations shown in columns two, three and four of Figure 1, provided that the collider has symmetric beams and no reason to choose any $\phi$-orientation more than any other.

To avoid the MNIST-based dataset having an unfair handle (orientability) not possessed by real jet data, this discrepancy needs to be removed. One way of removing it would be to augment our dataset by adding $y$-translations of uniformly random magnitude to every image in an LHC dataset (being sure to wrap pixels lost from the top back onto the bottom) before then giving every image a 180 degree rotation with probability 50%.

Although one could augment the dataset in that way, it is more efficient to simply make the neural net invariant with respect to those transformations. Doing so is as easy as making the net parity-odd, the only difference being that one seeks symmetry rather than antisymmetry and so one must combine sub-nets with symmetric operations (like maxpooling or summing) rather than antisymmetric operations (like subtractions).

## 4  An example parity-odd axisymmetric network

Recall that the necessary axisymmetry comes from establishing (i) an invariance with respect to $y$-translations of 28x28-pixel images corresponding to $\phi$-invariance in the real world, and (ii) invariance with respect to 180 degree rotations of the 28x28 images corresponding to interchange of beams and isotropy.

Our final network $f(x)$ gains its parity-oddness and one-half of its axisymmetry (the part requiring invariance with respect to 180 degree rotations) by being defined in terms of a sub-network $g(x)$ as follows:

$$f(x) = g(x) + g(R_{180}x) - g(Px) - g(R_{180}Px). \tag{1}$$

The sub-network $g(x)$ (implemented in pytorch by Paszke et al. (2019)) therefore only needs to respect the $\phi$-rotation (or $y$-translation) invariance. The $g(x)$ sub-network starts by having two 2D-convolutional network layers (one with kernel size $4 \times 4$ in $\eta \times \phi$, and the other with size $14 \times 6$ in $\eta \times \phi$, always stride 1) to provided basic image processing of the 28x28 pixel images fed to it. These two layers bring the number of features up to 32 and then to 64, and have the appropriate over-run regions to allow them to implement the required periodic boundary conditions. They are then followed by a max-pooling over the $\phi$ direction to finally achieve the desired $\phi$-invariance. $g(x)$ then concludes with two fully connected layers reducing 640 features to 128 and thence to a single feature so that $f(x)$ can then make used of $g(x)$ in the manner outlined in (1). To describe the structure of our network, we annotate its printout (as a pytorch Module) as follows:

```
Net(
  # phi-cyclic pad
  (conv1): Conv2d(1, 32, kernel_size=(4, 4), stride=(1, 1))
  # relu
  # phi-cyclic pad
  (conv2): Conv2d(32, 64, kernel_size=(14, 6), stride=(1, 1))
  # relu
  (max_pool): MaxPool2d(kernel_size=(28, 2), stride=(28, 2), ...)
  # flatten
  (fc1): Linear(in_features=640, out_features=128, bias=True)
  # relu
  (fc2): Linear(in_features=128, out_features=1, bias=True)
)
```

The network is trained on images provided in batches. Each batch

$$B = \{x_1, x_2, \ldots, x_{|B|}\}$$

contains $|B| = 64$ images. For each batch the mean net score $\mu_B$ is calculated:

$$\mu_B = \frac{1}{|B|} \sum_{x \in B} f(x).$$

The standard deviation of the scores of the images in the batch is also recorded as $\sigma_B$. The net is trained to maximise $\mu_B/\sigma_B$. This process encourages the net to do its best to give every image a common constant positive value, and to eschew negative values if possible. The net is trained to maximise $\mu_B/\sigma_B$ rather than $\mu_B$ alone since the former (unlike the latter) cannot be trivially made larger by replacing $f(x)$ with $\lambda f(x)$ for some constant $\lambda > 1$. The complete code for the network may be found in Lester (2021b).

Though it has already been emphasised, we repeat the message that the model chosen above for $g(x)$ and the choice made for the loss function $\mu_B$ are almost certainly non-optimal. Indeed, that is one reason why the more detailed paper exploring realistic particle physics events (Lester et al. (2022b)) tests a variety of very different models and very different loss functions. None of that lack of optimality matters for the purposes of the current paper, however, whose role is pedagogical. The only things that matters in this paper are the way that $g(x)$ has been (anti)symmetrised into $f(x)$, and (as will be seen quantitatively in the results section) that the mechanism works despite its simplicity.

## 5 Results

Quantitative results are presented in Table 1 and confirm that parity violation is not detected where it should be absent (the most important result) and yet is discovered where it is present (the bonus).

The same results are presented qualitatively and graphically in Figure 3 in which the preponderance of green in plots (a) and (c) within Figure 3 shows that the networks trained on the chiral modes are able to label the majority of their inputs with positive scores according to $f$, meaning that they have identified that these datasets are chiral (i.e. violate parity). In contrast the nets have not managed to achieve this for the achiral controls ((b) and (d) in the same figure), despite having been trained on achiral control data. It is notable and pleasing that the networks are not flummoxed by the differences between mode-1 and mode-2.

| Mode | $\rho^+_{\text{chiral}}$ | $\rho^+_{\text{achiral}}$ |
|---|---|---|
| mode–1 | 90±1 % | 51±2 % |
| mode–2 | 89±4 % | 42±6 % |

Table 1: When tested on groups of 1024 images in the relevant test dataset, the above numbers show the typical fraction $\rho^+$ of images per batch which were assigned a positive value by the function $f$ after training. As desired, the chiral LHC datasets are flagged as chiral at a statistically significant level, while those LHC datasets which are achiral are not, with their fractions remaining consistent with the 50% expected by chance alone.

Figure 4 demonstrates that our network function is indeed parity-odd and axisymmetric as desired.

## 6 Discussion.

The strategy has achieved its goals. What is perhaps more interesting to reflect on is what this buys in comparison to other methods for analysing multi-jet events for evidence of parity violation in colliders (such as the Large Hadron Collider) without polarised beams.

Only one work (Lester et al., 2022a) has previously tried to tackle this task in a systematic way. To give a chastening example: when that work considered collisions of the form $S_{ab \to jjj+X}$ (three important jets, and perhaps some other stuff $X$ in the final state, resulting from the collision of an 'a' with a 'b') it was found that 19-dimensional continuous ideal vector parity (to use the nomenclature of Lester (2021a)) would, in principle, detect any parity-violating signature. However: (i) over 60 pages of algebra were necessary to

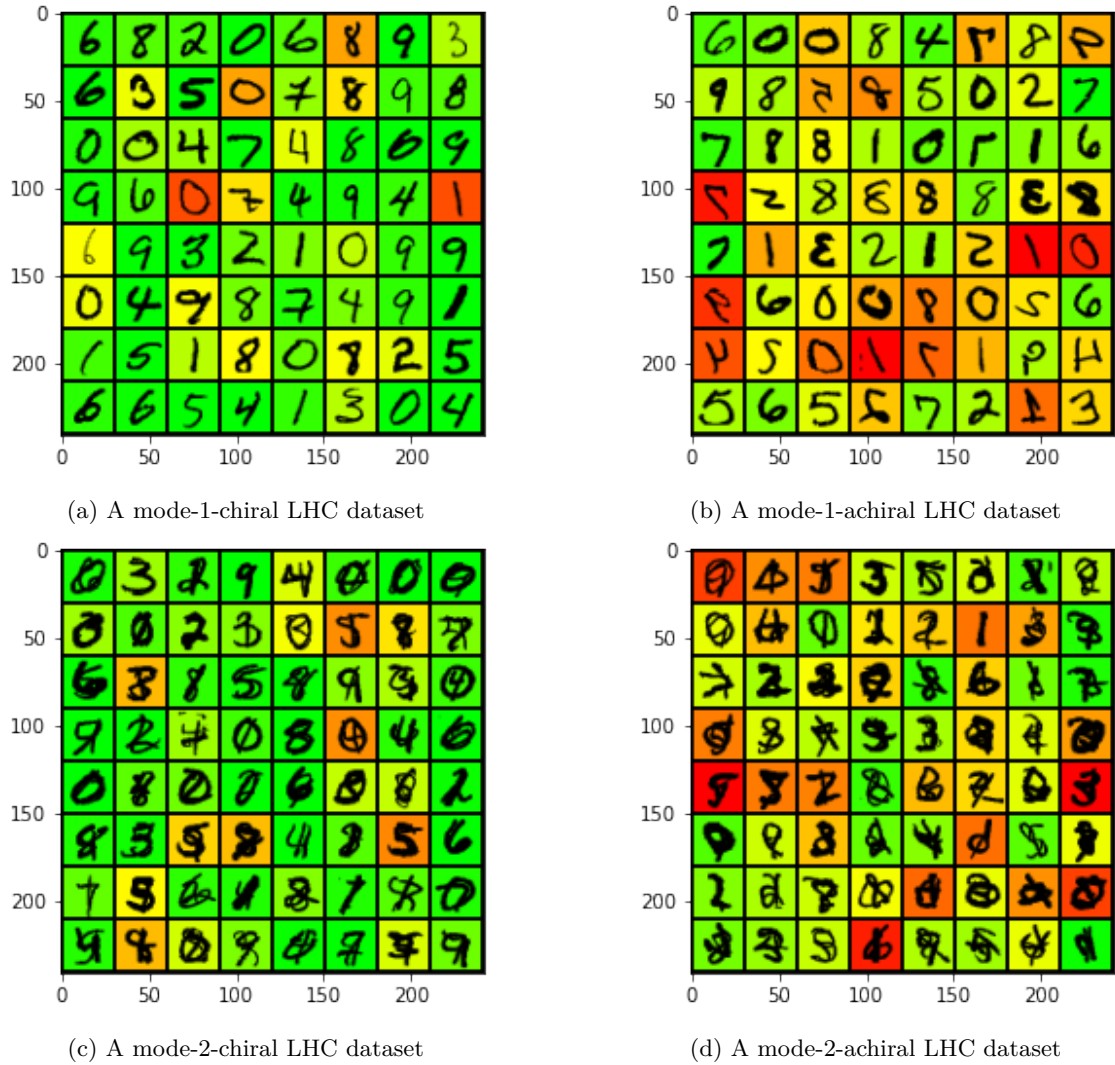

(a) A mode-1-chiral LHC dataset

(b) A mode-1-achiral LHC dataset

(c) A mode-2-chiral LHC dataset

(d) A mode-2-achiral LHC dataset

Figure 3: Four types of LHC dataset from the test dataset are shown after having been coloured according to the output of the net. Bright green represents increasingly positive numbers. Shades of increasingly vibrant red represent increasingly negative numbers. Yellow represents zero.

generate that result, and (ii) there seems little prospect of generalising that result to four final-state particles (due to factorial increases in computation complexity) let alone to $n$ final-state particles.

Why do we mention these limitations of Lester et al. (2022a)? The reason is that the present work can be considered to be doing a very similar job to Lester et al. (2022a) but for events with 'up to' 784 = 28x28 jets in the final state, yet in a way that scales much better.[10]

---

[10]The intensity of each pixel can, in principle, represent a very narrow jet, and so an image with (say) only three black pixels could represent a three-jet event. Moreover, the permutation symmetry modelled between jets in Lester et al. (2022a) carries over to our images, since the pixels to not somehow list jets in a particular order. The one major difference between Lester et al. (2022a) and the present work is, however, that Lester et al. (2022a) provides guarantees of ideality (see definition of this term in Lester (2021a)) which the present work never can.

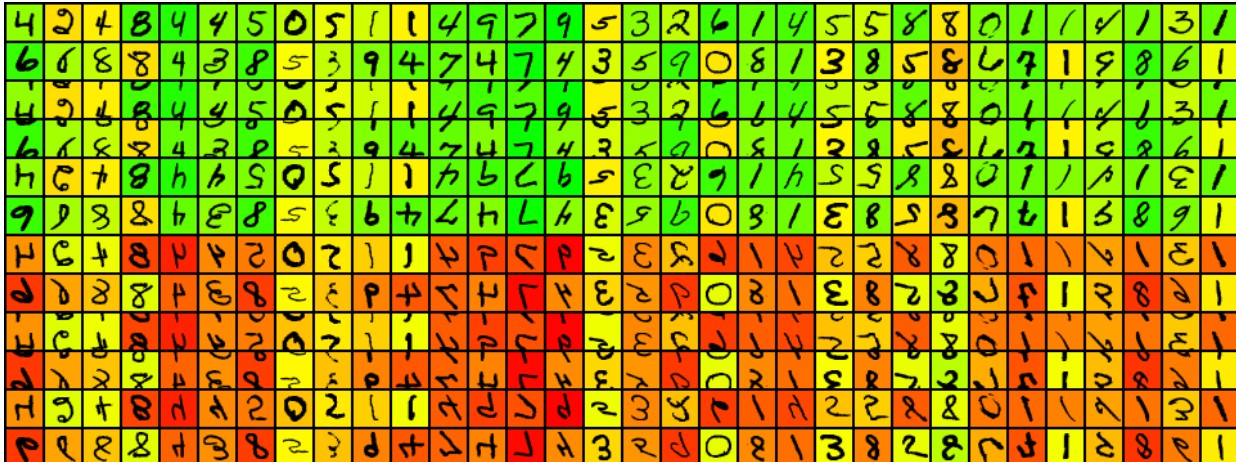

Figure 4: Here the same 64 letters from a mode-1-chiral LHC dataset are evaluated by the same symmetrised network $f(x)$ in six different ways. In the first pair of rows (the control rows) the images are evaluated without any changes at all. Most of the controls are coloured green (as one might expect) but the fact that this happens is irrelevant for the purposes of the check being performed here. What matters is how the colours in those two rows relate to the colours in the remaining rows. In the next pair of rows the nets are re-evaluated but after a shift (or roll) down of each image by 10 pixels. In the next two rows the transformation prior to evaluation is instead a 180 degree rotation. The key take home message from these first six rows is that (as the colours do not change!) all of these transformations do not affect the network score, and so the network is axisymmetric as desired. This property is not learned, it is guaranteed by construction. Nonetheless, it is nice to see a graphical check that the implementation does what was expected. The lower six rows repeat the upper six rows, but after an initial left-right parity flip of each image. In this case we see that there is (as desired) an immediate flip in the output from positive to negative (or in rare cases vice versa) because the network is, by construction, parity-odd.

## 7   Conclusion

This paper has proposed a framework for conducting tests of symmetry violation at the LHC. A key feature of our proposed methodology is that it can be applied to recorded data: a discovery would be made by comparing collision events to themselves rather than to theoretical predictions (of which there are few). The proposal therefore plugs a gap in the literature by making discoveries possible which hitherto could easily have been missed on account of the lack of theoretical models resulting in a lack of dedicated searches. This is important: the LHC is exploring centre of mass energies which are orders of magnitude higher than those in which parity violation was first discovered, and there is therefore no knowing what mysterious signals could be hiding in LHC datasets created in the last decade. Important signals could be hiding in plain sight.

This paper is intended to serve a primarily pedagogical and awareness-raising purposes. It is also intended to approachable to readers outside of particle physics. Accordingly it was decided to demonstrate the viability of proposed methodology by applying it to a toy problem wherein the hidden parity violating signature was the handedness of human handwriting. This is a simple target symmetry (parity) in an idealised detector possessing equally simple 'unimportant' intrinsic symmetries (axisymmetry and beam interchange symmetry) which are reminiscent of symmetries found in detectors at particle physics colliders.

A more complex sister paper (Tombs & Lester, 2022) describes how the framework presented here could easily be extended to consider continuous symmetries and shows how to deal with important issues (notably acceptance and inefficiency) which were ignored here for reasons of accessibility. A more 'realistic' particle physics demonstration is developed in a sister paper Lester et al. (2022b).

## 8 Acknowledgements

The authors greatly acknowledge discussions with Ben Nachman related to the present work and to Collins et al. (2018), and with Shikma Bressler related to the present work and to Volkovich et al. (2022). Useful discussions with Radha Mastandrea, Daniel Noel and Vidhi Lalchand are all very much appreciated, as has been the patience of Tom Gillam in assisting with numerous questions about pytorch. Finally we note that the paper was considerably improved as a result of open-review discussions with three anonymous referees and one editor in the TMLR journal. We thank them for their time and attention.

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
