# OpenReview forum: "Using unsupervised learning to detect broken symmetries, with relevance to searches for parity violation in nature."
_TMLR — Accepted by TMLR_

### Review · Reviewer_ghxN · 2022-09-09

**Summary Of Contributions:**

The paper presents a machine learning method for detecting symmetry violations in data. The approach is general, but motivated by an imporant problem in high-energy physics. The method is demonstrated on a toy problem.

**Broader Impact Concerns:**

None.

**Requested Changes:**

None.

**Strengths And Weaknesses:**

Strengths:
- The writing is exceptionally clear. Figure 1 is excellent. The paper was a joy to read!
- This is a clever idea and could prove very powerful. I am not a physicist and don't know all the details of the application, but I see no flaws in the argument here. The approach has many advantages over the existing approaches to testing for parity violations (namely, looking for unexpected "bumps" in histograms, or coming up with alternative hypotheses and performing Monte Carlo simulations to test each one).
- The discussion makes a good argument for why this ML approach is valuable.
- While the motivation comes from HEP, this is an interesting ML strategy for data analysis, and could potentially be useful for data analysis in other fields. It makes sense to publish this work in an ML venue.
- The authors use a toy dataset to demonstrate their method, and refer to a separate paper which applies the method to particular application in HEP. I think this decision makes sense.


Weaknesses
- None.

---

### Review · Reviewer_jRVr · 2022-09-09

**Summary Of Contributions:**

The authors propose a method to spot parity violations using parity-odd functions. The motivation comes from high energy physics.

**Broader Impact Concerns:**

No broader impact concerns.

**Requested Changes:**

For the paper to be fit for publication in TMLR, some possible suggestions could be to try to incorporate more technical details; investigate (theory, experiments) the difference between trained and untrained networks and discuss assessment of statistical significance for both cases; extend the work to symmetries different from parity; discuss aspects of practical applicability of their method to LHC data or present any potentially interesting application to any other data. It may well be that a paper with the aforementioned changes ends up being very close to one of the "sister papers".

**Minor:**

The Pytorch pseudocode for the neural network appears unnecessary.

In section 2, the authors use the term "overtraining": presumably they mean "overfitting" (the latter term is more standard in machine learning)?

I would consider modifying the title: the joke is a bit of a stretch given no serious references to GANs in the paper.

**Strengths And Weaknesses:**

First, I want to stress that I do not know much about machine learning for high energy physics, and my review should be weighted accordingly.

**Strengths:** The paper is written in a lightweight and simple way.

**Weaknesses:**

1. Are the claims made in the submission supported by accurate, convincing and clear evidence?

Claims in the submission are mostly qualitative and nontechnical. For any more rigorous treatment, or theoretical results, the authors refer the reader to multiple "sister papers". Key concepts are never formally defined. The presented experiments are mostly qualitative.

Take for example section 2: _"In practice, of course, the training step is very important. Without training there is little chance that the net will actually be attending to any useful features of the data [...] the training process exists [...] only to increase the probability of being able to claim discovery"_. No further elaboration on this paragraph is presented anywhere in the paper: In what sense should it be important for the neural network (or other function approximator) to _"attend to any useful features of the data"_? How does the _"probability to be able to claim discovery"_ change when the deployed neural network is trained, as opposed to when it is not? Can you show, in theory or experiments, this difference in performance of trained vs. untrained models?

The authors do not provide any formal definitions in the paper. For example, in section 1, the authors present one of their main ideas _``that a parity-odd function $f(x)$ can only have an asymmetric distribution if the dataset on which it has been evaluated contains some elements $x_1, x_2, \ldots$ which differ in frequency from their parity-flipped partners $Px_1, Px_2, \ldots$"_ While this sentence is probably meant as a nontechnical exposition of the main idea, no technical exposition follows. The authors do not explain what is the domain of the random variables, whether they are real-valued or categorical, what is $P$ and how are the parity-flipped partners defined etc---all of this is left to the imagination and intuition of the reader.

The assessment of statistical significance of the discovered parity violation (and how and whether it differs between trained and untrained networks) is not discussed. The only reported quantitative output of the experiments is table 1.

2. Would some individuals in TMLR's audience be interested in the findings of this paper?

A strong motivation of this work seems to be the analysis of LHC data. At the same time, according to the authors, the manuscript aim to expose some very simple ideas, abstracting away the complexities of the high energy physics application as much as possible. In the words of the authors, their aim is to _"provide a simple and pedagogically useful illustration of the key message, free from other distractions"_. For any theoretical results, discussions of complexities coming from applying the method to LHC data and extensions to symmetry other than parity readers are referred to multiple "sister papers".

The question is whether what is left is interesting enough, and whether such pedagogical aim makes this work appropriate for publication in TMLR. Overall, my impression is that the effort towards abstracting away details of the LHC application might have gone too far, ultimately leaving too little technical content.

Let me further elaborate. This paper is clearly not about GANs, as the authors candidly admit in footnote 4. The authors also argue (footnote 6) that the paper is not really about neural networks either; and it could even be said that it is not about (machine) learning, based on section 2, paragraph "Unusual Features of This Proposal". Ultimately, it is unclear what aspects of the paper should be interesting for the TMLR readers.

It is well possible that someone interested in the high-energy physics applications might immediately recognise the value of the work (I myself am admittedly not, therefore I am unable to answer the question:_"Is there something to be learned by some researchers in their area from their work?"_): in which case, one wonders whether this paper is at all necessary, and whether it wouldn't instead be better to present this ideas to the high energy physics community in a dedicated, subject-specific workshop or venue.

---

> ### Author Response · Authors · 2022-09-09
> **Author response to comments of reviewer jRVr    (3 of 3)**
>
> RRR : The question is whether what is left is interesting enough, and whether such pedagogical aim makes this work appropriate for publication in TMLR. Overall, my impression is that the effort towards abstracting away details of the LHC application might have gone too far, ultimately leaving too little technical content.
>
> >>> : The referee and TMLR editors are entirely within their rights to determine what is in-scope.  We think TMLR is a good place for a paper like this. One reason is that given in the first few paragraph of the paper: namely that despite producing 1000s of papers the LHC collaborations have lost sight of a simple fact: that broken symmetries can be spotted in very general ways. If LHC collaborations of tens of thousands of particle physicists can lose sight of (or fail to grasp) an important idea like this, then it is almost certainly the case that scientists in other fields are making similar mistakes. Therefore there is (we would argue) a role to play in attempting to publicise useful ideas, even if they are simple at heart, and especially if the benefits of using those ideas is great and their utility has not been appreciated by a large user base. Communication works best when papers are simple, not complex, hence the emphasis on simplicity here. Another reason is given a the end of this comment.
>
> RRR : Let me further elaborate. This paper is clearly not about GANs, as the authors candidly admit in footnote 4.
>
> >>> : As the referee will have seen, the word GANs in the title is just a joke to make the title of a paper about mirror-symmetry palindromic and thus memorable.  If TMLR editors don’t like word-play the title can be made drier and joke-free.
>
> RRR : The authors also argue (footnote 6) that the paper is not really about neural networks either; and it could even be said that it is not about (machine) learning, based on section 2, paragraph "Unusual Features of This Proposal".
>
> >>> : Absolutely not.  The above statement is a mis-reading of the paper caused by the referee appearing to believe we have said somewhere that our nets don’t need to be trained.  Unquestionably the training step (as explained in the first para of the sec the referee mentions) is needed in practice.  The simple reason is that broken cups don’t reassemble themselves and neural nets (or any other ML algorithms) are useless if you don’t train them — unless you are willing to look for successes which have only infinitesimal probabilities of happening. Training is necessary, and ML alg architectrure very important, and that is why the ML community has a strong role to play in making any actual attempt to use our approach useful in practice.  Our earlier suggestions of ways we could improve the words which have mislead or confused this referee will, we hope, remove the one and only source of this suggestion that our work is not about machine learning.
>
> RRR : Ultimately, it is unclear what aspects of the paper should be interesting for the TMLR readers.
> It is well possible that someone interested in the high-energy physics applications might immediately recognise the value of the work (I myself am admittedly not, therefore I am unable to answer the question:"Is there something to be learned by some researchers in their area from their work?"): in which case, one wonders whether this paper is at all necessary, and whether it wouldn't instead be better to present this ideas to the high energy physics community in a dedicated, subject-specific workshop or venue.
>
> >>> : Ultimately it is for TMLR to decide what its remit contains. We ourselves were introduced to TMLR by someone who highlighted what they claimed was a key TMLR philiosophy and described on the TMLR home page as follows:
>
> 	“TMLR emphasizes technical correctness over subjective significance, to ensure that we facilitate scientific discourse on topics that are deemed less significant by contemporaries but may be important in the future”
>
> >>> : The above statement was music to our ears, chiming with exactly the vision we had for this paper. Our paper seeks to make simple idea concrete, an idea that certainly needs using by LHC collaborations (whose members will benefit from having access to a simple/toy exposition and use-case) since attempts to discover new sources of parity violation have not featured once in the first 1000 papers not through lack of interest but because collaborations can’t see technically how to do such searches.  That is despite parity violation being the most important discovery of the 1950/1960s.  By extension one assumes that researchers interested in discovering symmetry breaking in other fields may be similarly stuck and could use the same help that his paper intends to offer people on the LHC. With ML (particularly ML training) being the crux of the technology needed to make this idea work in all practical circumstances, TMLR, with its philosophy on correctness and discourse in machine learning, seemed to be the ideal home.

---

> > ### Comment · Reviewer_jRVr · 2022-09-12
> > **Reply to the authors**
> >
> > Thank you for your reply. Some further comments below.
> >
> > I want to stress that I cannot judge on the significance of the idea in the context of high energy physics. I only want to mention that machine learning aspects are mostly abstracted away in the submission. At the same time, in their reply, the authors say that
> > > ML (particularly ML training) [is] the crux of the technology needed to make this idea work in all practical circumstances
> >
> > Given this, it might be beneficial to discuss training and its role in any implementations of this proposal in a more detailed manner, and possibly more quantitative. For a researcher interested in exploiting the idea presented in the manuscript it may otherwise be unclear what (machine learning) problems they may encounter when applying the proposed idea (e.g., say a network does not output a nonzero mean: is it because it is not a _"good net"_, or because it was not trained properly? How is a researcher to know?), let alone how to go about solving them. I wonder whether adding a paragraph or subsection on limitations and foreseeable issues when applying this idea in practice might be beneficial (some comments on this are present at various places in the text, often pointing to sister papers, but maybe a dedicated section or paragraph could be useful?); and whether displaying an application of this to LHC data, and discussing the technical difficulties of applying the idea in that context, could help make the benefits of the proposed approach, as well as its potential limitatations, more tangible.
> >
> > > TMLR emphasizes technical correctness
> >
> > One issue is that technical correctness may be hard to assess for qualitative statements. As an example, the authors say (in the manuscript):
> > > In practice, of course, the training step is very important.
> >
> > and (in their reply to my review):
> > > Unquestionably the training step [...] is needed in practice.
> >
> > These statements may sound reasonable, but they are rather qualitative: can readers rely on something else than opinion or intuition when judging their correctness? Some empirical evaluation (e.g., showing how ability to spot parity violation increases through training),  or any argument not requiring qualitative assessment, could help clarifying this. For another example, in section 6, the authors say that _"Figure 4 demonstrates that our network function is indeed parity-odd and axisymmetric as desired"_; but Figure 4 is visualisation and can only support a qualitative assessment.

---

> > > ### Comment · Reviewer_ghxN · 2022-09-13
> > > **Support for publishing the manuscript as-is**
> > >
> > > I agree with the authors' responses, and wanted to lend some additional thoughts.
> > >
> > > - Distinguishing between competing hypotheses in HEP typically relies on measuring a particular quantity that can be observed in experiments, and accumulating evidence for one hypothesis over the other under well-defined statistical models. That quantity might be observed directly, or it might be an "engineered" variable. Physicists commonly write papers on such variables that have been carefully engineered to separate hypotheses of interest. Machine learning has provided a mechanism to automate this process. The paper under consideration takes this a step further and suggests something similar to a "frequentist test" which enables us to reject a null hypothesis without specifying an alternative. This is a great idea and could almost certainly be used in other fields, so I appreciate that this manuscript is written for a general machine learning audience.
> > > - ML is certainly the crux of this approach. It relies on training a ML model to automatically identify asymmetric features of the data, which would otherwise be done by a physicist using extensive domain knowledge and through laborious experimentation.
> > > - I would like to push back on the comments by the other reviewers that more details are necessary. I think the idea of the paper is presented in a particularly clear manner, and there is a danger that adding details to satisfy review comments only obfuscates the main ideas by making the paper longer.
> > > - I was initially confused by the title, but I like it and think it is memorable.

---

### Review · Reviewer_7yco · 2022-09-10

**Summary Of Contributions:**

Parity violation is one of the phenomena which Large Hadron Collider is targeting to resolve. Particularly the main methodology right now for scientific discovery is to analyse jets and investigate particular decays and search for particular theory for parity violation. With that it is very hard to make progress quickly as we need to test and prove every theory and family of decays. Current paper propose hypothesis testing on violation of parity without making any prior on source / type of violation by training parity-odd approximation with maximization mean / std value computed on the batch and checking if this trained function can produce same-mean on test data. In case the predicted mean is shifted then the parity violation happens and we can claim discovery / problems with detectors symmetry. The paper consider pedagogical example on MNIST data modified with incorporating similar symmetry properties as in calorimeters and demostrating that we can use proposed method of hypothesis testing to detect which data has parity violation.

**Broader Impact Concerns:**

No any concerns as it is about particle physics which involve only nature questions and not related to any user sensitive information or user related predictions.

**Requested Changes:**

Overall idea itself is cool, that probably we can use simple unsupervised real data training and check if we have any parity violation. Some application may be if we just need to test that our simulated data for example have necessary properties. Or in case of real data (some known decay) when we know that it should be without parity violation but we see it - then we can recalibrate detector or estimate necessary systematic errors for future use. But this should be clearly stated in the paper. Right now it seems authors try to tackle more general problem of working with both detector and data parity violation. And moreover very toy, not physics related example is used (but data are very-very different in MNIST type and what we have in physics from my experience on working with both of them).

First, I would like to see authors to clarify the weak point I made above, discuss them and point to either my misunderstanding or uncover more precisely the procedure and more experiments.

Second I have several questions which would be nice to clarify in the text (especially for the part where I am wrong so that it is clear for future readers):
- maxpool and conv layers to have proper symmetry properties should have proper cyclic padding. Probably authors did this as I see in comments on code similar mentioning. But could you confirm that?
- how with proposed method we can distinguish / detect between parity violation in detector or in physics events as in real applications we will have definitely detector parity violation and maybe parity violation for physics?
- parity violation effect could be very small (correct me here, maybe I am wrong).  E.g. B-meson oscillations have very little probability and the methods should be very sensitive to this. So what is the limits of the proposed method with respect to this sensitivity? Could be that for real case scenario it will not work? In the past my team tried to use Mann-Whitney test (ROC curve) to detect if we have very rare decay in the real data - and we fail to use it as it was giving huge std, and effect we are searching is very little / very rare to be caught by models and this test.
- as I pointed above systematic errors from detector could be large and the reason why we have all these complicated not e2e methods is because we need to estimate properly all systematic errors. And exactly this is a root of problems. We use for example control channels to be able to calibrate ourselves and estimate these errors. If we have particular theory and particular decay study / energy band we better know what factors can affect so we can estimate them. In general case we cannot estimate all systematic errors. How do authors deal with this in proposed method? Also most probable we want to exactly estimate the violation level - and the proposed method even with detecting there is violation will not help to detect exact violation or help somehow to approximate it (correct me if I am wrong)
- what is the pdf of learnt values on train and test set for model output? how do we depend on data set size in train and test to be able to detect parity violation?
- different particles could probably have different level of parity violation - again question to sensitivity - for what kind of data method could be sensitive enough?
- the example considered in the paper is very simple and has 28x28 images. What is the actual calorimeter images we expect in different LHC experiments? How do we use other information from the event (e.g. full even reconstruction and jet properties estimation, like momentum)? As I mentioned in LHCb calorimeter images are very small in terms on number of pixels, so it could be that no any helpful information can be on that level to see parity violation.
- title is very misleading. I would prefer not to put GAN in title as it is creating more hype around them in physics. Right now GANs are tried actively to use for Monte-Carlo simulation which I don't trust much honestly. Also it is  better not to force people by title and abstract believe that GANs are able to solve problems of parity violation.


**Strengths And Weaknesses:**

**Strengths**
- idea on hypothesis testing with model training without any prior on the parity violation source / type / theory, working with unlabeled **real** data
- nice analogues and examples in paper for people outside the LHC and physics community.
- probe the method on toy MNIST modified data

**Weaknesses**
- The main weak point is that no any test performed on real LHC data, even for known decays / events to demonstrate how it is working in real scenario. Or at least on simulated data with idealized detector or with detector with some asymmetries which we know.
- It is discussed a bit in the paper, but from a different view: we can detect the problem with detectors with this hypothesis testing. It is true, but then we have problem in real LHC experiments that we cannot detect and recognize if parity violation comes from new discovery or from detector systematic errors (which is more likely) and the hypothesis testing itself is not useful at all. We can spend years on proving that there is no any other systematic error but maybe we again forgot to account something. There is no any discussion or solution proposed in the paper on this.
- With both above points there is no any investigation how we depend on these detector asymmetries, how different scenarios of parity violation level and systematic error will affect each other for proposed method to claim discovery. There is no any evaluation of sensitivity of this hypothesis testing and which level we could expect to capture with it.
- I remember that at LHCb for example we had only 9x9 images of calorimeter (so very small compared to what discussed in the paper). It could be that for this level of images the method is not gonna to work. Also we have a huge amount of other information about the events which we can use and here it is ignored and not discussed how to use.
- The method proposes to maximize mean/std of output model values on a batch. In this case model can simply learn the constant (say 1, and then mean is 1 and std 0), e.g. either memorizing all the data or just learning to predict constant everywhere. In both case this function is useless and will not do the proper job on test set. How can we prevent from this and make sure we are doing the right job?

---

> ### Author Response · Authors · 2022-09-14
> **First response to referee 7yco**
>
> Your "summary of contributions" regarding our paper is very accurate: both more complete and more succinct than I could have managed myself.
>
> Although I will make a later response in which I reply on every one of the comments you have made, here today I think the first preliminary response I should give is to the "main weakness" you identify.  You say:
>
> >      "The main weak point is that no any test performed on real LHC data, even for known decays / events to demonstrate how it is working in real scenario. Or at least on simulated data with idealized detector or with detector with some asymmetries which we know."
>
> Our main response to the above comment is that we believed the right place to do all those important things was in a separate paper that would only/primarily be of interest to Particle Physicists (PP).  That separate PP paper is, in fact, already written, and although it was written AFTER the paper being reviewed here in TMLR, it is now already published in JHEP (a Particle Physics journal).   It may be found at http://dx.doi.org/10.1007/JHEP08(2022)231 or at https://arxiv.org/abs/2205.09876 [Note: the original posted arXiv link was wrong here. Now corrected!] .  The reason for wanting to write two papers was not to get "two publications for the price of one" -- an approach I dislike intensely -- but rather because we honestly believe that very simple (and yet unused) ideas which have the potential to be useful in contexts which are much much wider than those in which they were originally motivated deserve to be written up in an approachable format which shields readers from having to read about all sorts of things which (we argue) get in the way of communicating a core idea.
>
> A perusal of the above PP-specific version of this paper will, I suspect, convince all reviewers of why the "general" idea is not best communicated by going to PP-specificity.  All the PP-specific paper *does* contribute (and this the main motivation for *its* creation) is assurance that the the proposed general idea does have the potential to work in the actual sorts of situations that Particle Physicists might want it to (at least in idealised/simulated form).  That's something of value to the PP community, but we argue it's something that is of no value to (say):
>
> > the naturalist or medic who is trying to work out whether her electron microscopy images of helicobacter have one handedness of spiral more than  another,
>
> or
>
> > the astronomer looking for evidence of anisotropy in the cosmic microwave background,
>
> or
>
> > the financier looking for evidence of deviation of pairs of bond-prices from some symmetry that economists expect them to obey.
>
> All the above people (we would argue) deserve to be able to read a paper they can look at and understand simply in terms of things they already think they understand -- like handwriting -- so that they can then see the relationship to their own field.  That is our strong motivation for wanting to keep this paper "clean", and for isolating the PP-specific hard-evidence elsewhere.
>
> I plan to give responses to others of your comments later. But it is possible that some of them may not be required if you end up agreeing with our approach explained above.
>
>
> [ In the interests of full disclosure I should explain how the PP-specific paper has managed to be published before the simpler general one under review here, despite being written afterwards and taking longer to write. The main reason is that we were keen to get on with writing the PP-specific paper as soon as the current general paper had been made public on the arXiv.  Because the PP-specific paper took considerably more resources (both in terms of raw computing and in actual scientific validation) no effort was initially spent attempting to get the general paper published. It just sat on the arXiv.  A few months ago we thought we should do something about it and sent it to the first ML journal google told us about (Springer Machine Learning - MACH). They wrote back almost immediately saying the paper was "not fit for [their] journal style".  This response didn't entirely surprise us as we had never attempted to publish there before (all our previous papers were all PP-specific) and we had no real understanding of what they wanted.  Thinking that perhaps we should go for a very general audience we tried RSOC-PROCA only to be also told there that the idea was a good one, but their readers would probably not be interested in this sort of topic.  Despite two rejections, neither had criticised the paper content, and one journal had actually liked the content a lot, but both deemed the choice of journal just "wrong". So at that point we talked to  colleagues who (unlike us) specialise in ML, and they recommended TMLR.  This is where we find ourselves now --- but the initial delay in attempting publication, followed by the two "not right journal" rejections, have resulted in the harder paper beating the simpler one to publication. ]

---

> > ### Comment · Reviewer_7yco · 2022-09-14
> > **Thought on the paper style**
> >
> > Dear authors,
> >
> > Thanks for the quick reply!
> >
> > I am very happy to see some general ideas and even abstracting it away from PP to make more accessible to public. This is very right direction and I only support it. But I think we should  be careful in the way things are formulated. As having both background in LHC data analysis and ML I think it is valuable to have a proper good reference to the main PP paper and simplifying things in the current paper. This is partially done in the current version but I think it should be improved to have abstract but still clear and correct formulations. Then people as me or PP background can get the ML idea and go into details to the main paper. But you need to formulate clearly the summary and outcome / tests from the main paper with physics. Maybe having some section to discuss that / cross reference - and add note to ML background people to skip this section as done in some books.
> >
> > Next, I need to check the paper you referenced to understand better how it is applicable and used. But even without it I prefer to see discussion and explanation how you gonna to deal with the case where we have systematic error from detector and data together as in this case I don't understand how your method is applicable. And I am a bit close to biology, where this could potentially also be used, but they have the same issues with "detectors" which introduce even much more systematic errors. This should be discussed more clearly in the paper.
> >
> > Finally I have main question with respect to ML experiments themselves and decision making with your method. Right now I don't understand why it is learning not simple constant and what happens in memorization regime. I think this is important questions which will be great to have some understanding as right now in ML we have a lot overparametrization models and from scientific point we need to be sure models are doing proper things and do not cheat. All we know that we can artificially get mass peak, so we need to control even ML to avoid this artificial mass peak.
> >
> > I propose to make following modifications:
> > - stay with the tone of simple abstracted away example as methodological way
> > - add in the end small section on general discussion how this is applicable in PP, with reference and maybe main summary that it was successful, or what problems appeared
> > - make cleaner statements and use cases with proper explanation how I should use the method in any scientific discovery (because I could imagine that I can have some other property, not even some symmetry property, and I can test it in the same way assuming that I can have network / approximation model which preserve some property or has it). Step by step. With exact limitations and assumptions. Right now I feel it is very messy in the text and several concepts were mixed together.
> > - finally add anything on hypothesis testing. In any probabilistic testing we have the confidence interval for the decision making. So how does this work in your method? Do you have it or not? Even in simple MNIST example. As we always deal with decision making in science. What is sensitivity, how it depends on data size? Any first observations will be helpful as there are a lot of ideas but very small amount of working ideas. I think even in TMLR we aim to consider things which potentially can work not just ideas themselves.
> >
> > Happy to further discuss on particular points!
> > Best.

---

> ### Author Response · Authors · 2022-09-14
> **One other short clarification ....**
>
> With apologies for sending answers in a disjointed fashion, the following question has a simple answer which might benefit from being shared first:
>
> > The method proposes to maximize mean/std of output model values on a batch. In this case model can simply learn the constant (say 1, and then mean is 1 and std 0), e.g. either memorizing all the data or just learning to predict constant everywhere. In both case this function is useless and will not do the proper job on test set. How can we prevent from this and make sure we are doing the right job?
>
> The answer to this is that it is the parity-odd construction of the net which guarantees that the only constant that an odd function (or in this case specifically a "parity odd" function) can output is zero.  E.g. an odd real function f(x) satisfies f(x) = -f(-x) and so f(x)=C (a constant) implies that C=-C and so C=0.  Our function f(x) is specifically parity odd, so its oddness is guaranteed by the construction seen in our equation (1). If I simplify our equation (1) by dropping the 180-degree rotation terms (which are not relevant for the explanation I am giving) I can re-write it here as (1a):
>
>     f(x) = g(x) - g(Px).        (1a)
>
> where P is the parity operator that "mirrors" x, i.e. it turns our event x into its mirror image Px, and where g(x) is the function that the ML algorithm has the freedom to vary.  Mirroring is its own inverse, i.e. PPx = x for all x (i.e. P^2=1). So our (1a) guarantees that:
>
>     f(Px) = g(Px) - g(x).      (1b)
>
> and so comparison of (1a) and (1b) reveals that f(x) = - f(Px), which is the statement that the final output function f(x) is parity odd, regardless of the nature of g(x).  If f(x) were to settle on a constant, therefore, the only constant it could settle on is zero for the reason already given, which is also the reason why sin(x) has sin(0)=0.  The only thing that the ML algorithms can fiddle with is g(x). They can set g(x) to any constant (even a non-zero constant). But whatever constant g(x) is set to, f(x) would be forced to zero by construction.    The final parity-oddness comes from the way g(x) is used to define f(x).   Equation (1) is thus defining all the statements about "baked in" parity oddness
>
> A different way of saying the same thing is that any network which has learned to map every item of a training dataset to the constant "1" would have tied its hands in such a way that it would have forced itself to output "-1" on every item in the mirror image of that training dataset. If then on subsequent evaluation on a test-dataset the outputs had a statistically significant bias toward "1" (or even toward -1) then that would show that the test dataset does not contain mirror events in equal number to un-mirrored events - which would be a valid discovery of parity violation in that dataset regardless of how g(x) acquired its final form.
>
> In some sense this parity-oddness of f(x) coupled to ML's  freedom to do anything it likes with g(x) is the "the big (simple) idea" of the whole paper.  The paper uses parity oddness as an example because of the PP interest in parity violation, but notes that a sister paper shows how different forms of antisymmetrisation can be used to looks for violations of symmetries that are more complex than parity.  The choice of parity to demonstrate the idea is because parity is a simple symmetry, yet still interesting and relevant to both the LHC and to (e.g.) handwriting.
>
> If the net does learn a constant for g(x) and thus leads zero for f(x), as the paper already points out, this leads to a failure to discover evidence for parity violation: it does not lead to a false discovery.  That's discussed in the "unusual features of this proposal" section of the paper, and (I think) in parts of the introduction.
>
> A different referee has commented that we did not define some terms well enough, and included our non-definition of our parity operator "P" in that criticism.  Among the criticisms I've seen in this review process, I think that's probably the criticism that I feel I most would like to address: it's no good our having a paper that attempts to use parity violation as an example if we don't do a better job of explaining what parity is and why, therefore, (1) is unavoidably parity odd.  Similarly, when I have presented this work in seminars, I have usually found it necessary to have at least three different slides all of which attempt to describe (in different ways) how the baked-in (anti)symmetrisation process manages to guarantee that the only asymmetries seen are real ones .... because no one description is guaranteed to "click" with every member of the audience. Some want to see an equation. Some want to see other things.

---

> ### Author Response · Authors · 2022-09-21
> **Reply to weaknesses in "Review of Paper369 by Reviewer 7yco" after posting new paper version**
>
> The list below is in one-to-one correspondence with (and in the same order as) the bullet point list of "weaknesses" in the referee's original posting:
> * Forward references are now given to the paper ( doi:10.1007/JHEP08(2022)231 ) which does the in-depth (non toy) LHC test -- even though it was written after the paper under review. Also, the whole paper's text is edited to better explain the benefits of hvcing a toy version exists in parallel with the more in depth one.
> * Everything the referee says in the second weakness bullet point is correct -- but the issue has no real answer. A set of bathroom scales measures the weights of things put on them, but if the readings on a set of scales go up over time, we don't know if the person who is being weighted on them should eat less or go on a diet, as it could just be that they are wearing more clothes as the weather is getting colder, or it could just be that the atmospheric density has gone down reducing their atmospheric buoyancy. The point I am making is that the "Mr Scales", the inventor of bathroom scales, did not have to justify how one could/should attribute weight measurements to actual increases in person-mass vs other effects. That's an important job, but it's not a job for the scales themselves, it's a job for the users of scales.  By the same token, the paper under review provides  a way of quantifying the degree of parity violation any  data stream it is fed, but it will always be up to the users of that information to decide what any signals they see are actually telling them about. We can't (and no one can) provide a single description of what they should do with the measurements.
> * In effect the reply to the last bullet point can appear again: how one deals with systematic errors and attributes observed effects to particular sources is all VERY IMPORTANT but is also VERY DOMAIN SPECIFIC. What needs to be done in this area will very much depend on what sort of data is being analysed, how it is being analysed, and what deficiencies is reasonable for a detector to have.  All very very important, (and likely 80% of any paper that would ever use a variant of the technique we are proposing), but none of it is addressable in the general case or without a specific real measurement in mind.
> * In https://arxiv.org/pdf/2205.09876.pdf we ended up being able to see parity violation in 32x32 calorimeter images where the source of PV was a BSM interaction that modified the qqg vertex. That's a big 'in your face' source of BSM PV. We would, I am sure, have been wholly UNABLE to see more standard sources of parity violation on account of the reason given in footnote 1 of https://arxiv.org/pdf/1904.11195.pdf -- which ties observable sources of SM parity violation only to things to the level of CP violation (which is indeed very small) -- but there is already a well developed literature looking for CP violation within the SM. Fortunately the work under review never claims to be able to see all parity violation that is present, it only claims to be able to have sensitivity to all observable sources of parity violation -- i.e. all those which "in principle" can be observed in a given machine given it's own quirks.  The more subtle is the source of PV, the harder one would need to work on data pre-selection and on input features in order to see things: tricky measurements always take the most time and there is no free lunch. But that's inevitable. The work under review does not ever claim to make PV discovery a fire-and-forget triviality.  It does not remove the need for physicists to select data with great care to feed to the analysis framework so as to use their hunches to reduce the complexity of the problem. But what it does offer is a means for them to let ML algs identify the most promising sources of PV inside any substream of events that they choose to feed to the algs.
> * We put in more text in the new draft explaining what happens if the net g(x) learns to associate a constant with all data -- showing that this does not (cannot) lead to false discovery of PV where there was none.  We have not yet (but do wish to, and will) make add a conditioning term to the sigma_B in the last para of section 4 to remove the possibility of an undiscussed divide by zero. TODO!
>
> A repy on the same referee's "Requested changes' will follow.

---

### Author Response · Authors · 2022-09-14
**Understanding what to do next (addressed to anyone, but particularly to referees who have commented so far)**

Despite a large number of years of experience of "conventional" (i.e. non-open) review, this is my first experience of open review.

Regardless of whether this paper ever gets published by TMLR, I can say that the experience of open-review is proving to be (genuinely) interesting and positive -- despite being quite different to what I am used to -- and despite my not having expected the style of review to make much difference. It is particularly interesting to see discourse between referees being facilitated (i.e. not just author <--> referee discourse), as well as more immediate contact between referees and authors.

Anyhow -- given that this is my first experience of this "open" review process I might need some guidance from some of you as to what happens next.  Instructions sent to me by TMLR on 10th Sept (which I confess I only read today) say I should "respond to the reviewers" ... and that "The reviewers will be using this time period to hear from you and gather all the information they need ...".  I would therefore be interested to know sort of response that reviewers would prefer to see from us authors.

Prior to reading those instructions my "default" assumption was that I should produce "rebuttal" style responses to every individual point raised by a referee of the type I generated a few days ago for one referee.. [That's what we have to do in the LHC collaboration internal review processes, though there the referees are not anonymous, and it is usually not the referees themselves but someone else who forms the final judgement.. ]. I could produce more of them for each referee post made.

However if there is a more collaborative way forward (e.g. if the expectation is that the authors should commit to specific changes that they WILL make (rather than talk about the sorts of changes which could be made, if any) or if some third form of response would be more useful, then please say. I would rather work "within the system" rather than fight against it or just misunderstand it.

In the absence of any special recommendations,  I will wait a day or two and then write a response to the post made by referee 7yco , in whatever way seems most sensible to me at the time.

I presume that (beyond giving thanks) there is not really a need for me to respond directly to reviewer ghxN since his/her reaction to our paper is self-evidently very close to that we were aiming for.

---

> ### Comment · Reviewer_7yco · 2022-09-14
> **Reply**
>
> Dear authors,
>
> From my current experience what I did with other authors is the following: we did paper discussion on particular points and in general how the paper can be improved, what things seems to be wrong or which things are needed to be highlighted more. For some points they answer questions and make modification to be clear in paper, for some provide more experiments or clarifications, etc. So this is online process and authors time to time resubmit revision after some round of discussions. By the end of discussion period there is final draft where changes are reflected in the text and reviewers can make a final decision.
>
> I am ready to do several passes over new revisions and discuss exact places where I think the paper can be improved.
> Hope this is helpful, and maybe others have different suggestions and experience.

---

> > ### Author Response · Authors · 2022-09-14
> > **Thanks.**
> >
> > Thank you. This is helpful.

---

### Author Response · Authors · 2022-09-20
**New version for referees to consider**

Having had feedback from some referees about how to adjust to open review, and having had some discussions with some reviewers about their initial comments,  we've taken the view that the next step should be for us to generate a new version of the paper which aims to implement ~90% of the improvements that (in our view) the referee comments suggest are needed.

We have made those changes already, and (technology permitting) I will upload two new versions to TMLR right after posting this message . The first to be uploaded will be the new version itself. The second will be a "diff" version which will show you more easily the changes I made (with new text in blue, deleted text in red, and unchanged text in black).  I believe you can see all previous versions I upload, so I hope you can see both. If not, let me know and I will try to find another way of sharing both as a single file instead of as two versions.

Note that it has been a balancing act between, one the one hand trying to make minimal changes to the "ethos" of the paper (in keeping with one referee's feelings that it was clear as submitted) and on the other hand the acceptance (thank you other referees!) that there were numerous places where we could have been much clearer than we actually were -- and where we may have given the false impressions about what was (or was not) important.

We believe that the new version of the paper is much better for the changes .... which (we feel) have allowed the paper to become clearer and better at explaining itself, while not acquiring a feature-creep of details that would be better in more detailed accessory papers.

I have no time to do so today, but I still intend to go back an respond to original referee questions which have not yet been answered, but with the new version in front of you so that you can see that the questions being asked are "already addressed" in the new version.

We have

---

### Author Response · Authors · 2022-10-15
**Final update concerning all pending "referee requested changes"**

I am about to upload two new versions. The first is NOT a "camera ready" proof. The second should be camera ready.

The purpose of the first "non-camera ready proof" is to hold both:

(a) paper text which is equivalent to that expected in the camera-ready proof, together with

(b) a PDF (added as "supplementary information") containing a set of written responses to each "requested change" mentioned by a referee and which was not previously completely addressed by a response in the open-review discussion to date..

The reason for two separate uploads is that there is not space in these text comments boxes to give complete answers to the referees in ascii only, so a PDF response was preferred. The only place I could see to include that was via supplementary information. We do not, however, wish the "camera ready" version to include such supplementary information. The two uploads is the workaround used.

---

### Decision · Action_Editors · 2022-10-13

**Recommendation:** Accept with minor revision

**Comment:**

I would like to point out that I perfectly understand the reservations of the two reviewers who suggested a lean reject (but who acknowledged the scientific value of the paper), and I thank them for their careful work which has lead to these improvements.  I think that appreciating the style of this paper is really person-dependent. But through my own reading, as explained before, I think that the conceptual content overweight the lack of technicality in this case.
The discussions have lead to great improvements of the paper but I would like to ask the authors to implement this change of title and, if possible, take another look at the comments of reviewers and potentially improve further, in particular concerning the relations to "real applications". Once these revisions have been implemented, the paper should be in a position to be published.

**Audience:**

I do believe that part of the TMLR audience will be interested by the main ideas of the paper, given their simplicity and generality. The paper is accessible to students too. But when I read first the title I really had no idea what it was about. I understand the comic aspect of the title a-posteriori, but I fear that potential interested readers will skip it due a mis-understanding of the title. I therefore ask the authors to provide a clear title truly related to the paper's content.

**Claims And Evidence:**

Two of the reviewers emitted doubts on the level of 'rigor' and (lack of) 'technicality' of this paper. Basically they argue that the lack of statistical test formulation and precise statements and definitions makes the approach hard to grasp and really use. This lead them to lean towards rejection. The third reviewer has the opposite viewpoint and sees the simplicity of the paper has an asset, allowing to convey a generic and potentially useful idea for more general scientific purposes than high-energy applications. The review is very positive and provides a clear accept. Given the deviation between these I read the paper in details. My perspective is that yes, it is true that the paper  stays at a very qualitative level, and the absence of formal definitions and more quantitative results can be disconcerting for some readers. But like the third reviewer I found the message overall very clear, and I believe that the paper has a high scientific value. I also got the impression that the simplicity of it gives a way to access quickly a beautiful and rather generic idea, and food for thought for other contexts. Readers interested by more details can dive in the more technical papers cited there, while having already a clear idea of the main concepts. I also think that in spite of its lack of formal statements, the claims are well supported by convincing experiments carried on this simple toy model, and can be reproduced if needed. So my judgment is positive on this point.

---

> ### Author Response · Authors · 2022-10-14
> **Interim response to action editor**
>
> On behalf of myself and my co-author, I would like to thank the "action editor" for his/her recent post, and am pleased to indicate that we are more than happy to adapt to its requests -- including both title change and continued edits based on pre-existin comments from other referees.
>
> Although it has been a few weeks since I posted here, that delay has been caused largely by the start of the university term, and not a willingness to continue to address suggestions made by other referees -- they are just still "in progress".  I am pleased that the paper can remain largely non-technical, but I also value the feedback from the reviewers who wanted the opposite, because there are many things which they pointed out (some of which were addressed in the last uploaded version, and some of which will be addressed in the next) which have made the paper much better already.  It may always be a bit of a "Marmite" paper (this being the UK-centric way of describing things which people tend either to love or hate without much middle ground) but it is nice to see that marmite papers can sometimes get published.